# Genomics reveals heterogeneous *Plasmodium falciparum* transmission and selection signals in Zambia
Abebe A. Fola[1,8,9], Qixin He [1,9], Shaojun Xie[2], Jyothi Thimmapuram[2], Ketaki P. Bhide[2], Jack Dorman[1], Ilinca I. Ciubotariu [1], Mulenga C. Mwenda[3], Brenda Mambwe[3], Conceptor Mulube[3], Moonga Hawela[3], Douglas E. Norris [4], William J. Moss [4,5], Daniel J. Bridges [6] & Giovanna Carpi [1,4,7] ✉

## Abstract

**Background** Genomic surveillance is crucial for monitoring malaria transmission and understanding parasite adaptation to interventions. Zambia lacks prior nationwide efforts in malaria genomic surveillance among African countries.

**Methods** We conducted genomic surveillance of *Plasmodium falciparum* parasites from the 2018 Malaria Indicator Survey in Zambia, a nationally representative household survey of children under five years of age. We whole-genome sequenced and analyzed 241 *P. falciparum* genomes from regions with varying levels of malaria transmission across Zambia and estimated genetic metrics that are informative about transmission intensity, genetic relatedness between parasites, and selection.

**Results** We provide genomic evidence of widespread within-host polygenomic infections, regardless of epidemiological characteristics, underscoring the extensive and ongoing endemic malaria transmission in Zambia. Our analysis reveals country-level clustering of parasites from Zambia and neighboring regions, with distinct separation in West Africa. Within Zambia, identity by descent (IBD) relatedness analysis uncovers local spatial clustering and rare cases of long-distance sharing of closely related parasite pairs. Genomic regions with large shared IBD segments and strong positive selection signatures implicate genes involved in sulfadoxine-pyrimethamine and artemisinin combination therapies drug resistance, but no signature related to chloroquine resistance. Furthermore, differences in selection signatures, including drug resistance loci, are observed between eastern and western Zambian parasite populations, suggesting variable transmission intensity and ongoing drug pressure.

**Conclusions** Our findings enhance our understanding of nationwide *P. falciparum* transmission in Zambia, establishing a baseline for analyzing parasite genetic metrics as they vary over time and space. These insights highlight the urgency of strengthening malaria control programs and surveillance of antimalarial drug resistance.

## Plain Language Summary

Malaria is caused by a parasite that is spread to humans via mosquito bites. It is a leading cause of death in children under five years old in sub-Saharan Africa. Analysis of the malaria parasite's complete set of DNA (its genome) can help us to understand transmission of the disease and how this changes in response to different strategies to control the disease. We analyzed the genomes of malaria parasites from children across Zambia. Our study revealed that 77% of children harbored multiple parasite strains, which suggests that local transmission (transmission between people within the same local area) is high. Genetic evidence for long-distance transmission was rarer. Furthermore, our findings suggest parasites are evolving in response to antimalarial drugs. Our study enhances our understanding of malaria dynamics in Zambia and may help to inform strategies for improved surveillance and control.

[1]Department of Biological Sciences, Purdue University, West Lafayette, IN, USA. [2]Bioinformatics Core, Purdue University, Purdue University, West Lafayette, IN, USA. [3]PATH-MACEPA, National Malaria Elimination Centre, Lusaka, Zambia. [4]The Johns Hopkins Malaria Research Institute, W. Harry Feinstone Department of Molecular Microbiology and Immunology, Johns Hopkins Bloomberg School of Public Health, Baltimore, MD, USA. [5]Department of Epidemiology, Johns Hopkins Bloomberg School of Public Health, Baltimore, MD, USA. [6]PATH, Lusaka, Zambia. [7]Purdue Institute for Inflammation, Immunology, & Infectious Disease, Purdue University, West Lafayette, IN, USA. [8]Present address: Department of Pathology and Laboratory Medicine, Brown University, Providence, RI, USA [9]These authors contributed equally: Abebe A. Fola, Qixin He. ✉e-mail: gcarpi@purdue.edu

Although progress toward malaria elimination has recently stalled, malaria control interventions have averted significant morbidity and mortality since 2000[1]. Surveillance is increasingly critical to sustaining progress toward malaria control and elimination by characterizing changes in transmission, maximizing intervention impact, and identifying threats to elimination. Traditional surveillance methodologies such as Malaria Indicator Surveys (MIS) can now be augmented with genomic approaches to provide additional information using more sensitive metrics of transmission intensity, including quantifying parasite population diversity in response to control interventions, as well as identifying genotypes associated with drug resistance[2,3].

Population genomic surveillance has been used to assess *Plasmodium falciparum* transmission dynamics and population structure during declining transmission[4], outbreaks[5], resurgence[4], and epidemic expansion[6], as well as to identify population differentiation and loci under positive selection[7]. Population genomic metrics such as low multiplicity of infection (MOI, the number of genetically distinct parasites of the same *Plasmodium* species within host), low genetic diversity, geographic clustering, and inbreeding with highly related parasite pairs by identity by descent (IBD) are expected in low transmission settings with declining transmission. High transmission intensity is associated with high levels of MOI, high genetic diversity, low parasite relatedness, and limited population structure due to extensive parasite recombination rates[8]. Identifying parasite population clustering and local heterogeneity has major implications for assessing the feasibility of targeted control approaches to achieve malaria elimination[9]. Moreover, determining the spatial scale of parasite relatedness and parasite population structure could help to identify "sink" and "source" populations and capture spatial variation in transmission intensity to facilitate malaria elimination[10,11].

In Zambia, despite intensified control interventions, *P. falciparum* malaria remains endemic with highly heterogeneous transmission and variable parasite prevalence at subnational levels, making elimination efforts challenging[12]. Despite a north-to-south transmission intensity gradient based on epidemiological data, the country is using similar control strategies across provinces, such as mass distributions of long-lasting insecticide treated mosquito nets (LLINs), annual indoor residual spraying (IRS), prompt diagnosis with rapid diagnostic tests (RDTs) and light microscopy, and treatment with artemisinin-based combination therapy (ACT)[13,14]. Understanding the genomic structure of parasite populations and measuring the degree of parasite genetic relatedness are essential to assess transmission dynamics and the dispersal of infections, and to glean insights into how parasite populations respond to selection pressure exerted by different control interventions[3,15].

Whole genomic analyses of African *P. falciparum* parasite populations delve deeper than traditional malaria epidemiological surveys, offering valuable insights into parasite transmission patterns within populations and their interconnectedness[7,16]. Despite these advances, a nationwide population genomic study of *P. falciparum* within Zambia has been lacking. Previous efforts have been limited to targeted molecular genotyping with restricted geographical representation[12,17–19]. Unlike standard genotyping, where only a small fraction of the parasite genome is used to infer transmission dynamics from genetic signals, parasite genome surveillance using whole genome sequence (WGS) data can provide deeper and unbiased insights into malaria transmission intensity, parasite population relatedness, the degree of parasite mixing between different epidemiological areas[11,20], and signatures of selection[7,21].

To address knowledge gaps and support malaria elimination efforts, we conducted a nationwide genomic surveillance of spatially representative malaria parasites in Zambia by performing WGS of 241 *P. falciparum* samples from all provinces using dried blood spots (DBS) collected from children as part of the 2018 Zambia National Malaria Indicator Survey. To further contextualize *P. falciparum* genome diversity sampled in Zambia within Africa, WGS data for 781 *P. falciparum* samples representing 5 countries (Democratic Republic of the Congo, Ghana, Guinea, Malawi, and Tanzania) from the MalariaGEN Pf3k database were analyzed and included

for comparison. Using high quality genome-wide single nucleotide polymorphisms (SNPs) data we determined: (I) within-host parasite diversity ($Fws$); (II) parasite population differentiation across Zambia and between other countries; (III) spatial patterns of parasite population connectivity; and (IV) evidence of genome-wide detection of genes under positive selection. This study provides a high-resolution map of *P. falciparum* genomic diversity, transmission dynamics, and parasite population connectivity in Zambia. Moreover, it offers fundamental insights into how the implementation of control programs and elimination efforts may impact these parasite populations.

## Methods
### Ethical statement
The parents or legal guardians provided parental permission for study participants and this study was conducted with the approval of the Biomedical Research Ethics Committee from the University of Zambia (Ref 011-02-18) and from the Zambian National Health Research Authority.

### Sample collection and selection
Samples were collected during the 2018 Zambia Malaria Indicator Survey[22] that used a nationally representative two-stage stratified clustering sampling strategy with approximately 25 respondents per cluster across 179 standard enumeration areas from all ten provinces in Zambia, with oversampling in high transmission provinces. For statistical purposes, during the MIS each district within a province was subdivided into census supervisory areas (CSAs) and these were in turn subdivided into enumeration areas (EAs). The listing of EAs had information on the number of households and the estimated population size. The number of households was used as a measure of size for selecting the primary sampling units (PSU) which were the EAs or clusters. Blood specimens from children younger than 5 years were tested by RDT, microscopy and PET-PCR[14] and an additional dried-blood spot (DBS) was collected for parasite whole-genome sequencing. De-identified DBS were stored individually in plastic bags with silica gel desiccant at −20 °C before being shipped to the Carpi Laboratory at Purdue University, where they were stored at room temperature with fresh silica gel packets. For this study, we included 459 PET-PCR positive *P. falciparum* samples from ten provinces for sequencing (Supplementary Fig. 1). The majority of DBS samples collected from three provinces with low malaria transmission (Central, Lusaka and Southern) were negative by PET-PCR as well as by RDT and microscopy[14] limiting the number of samples that could be sequenced from these three provinces. DBS were registered and tracked in a database, where location, date of collection, and other metadata were recorded. Genomic DNA (gDNA) was extracted from single DBS samples using high-throughput robotic equipment (Qiagen QIAcube HT instrument) with QIAmp DNA 96-well kit according to the optimized high-throughput gDNA extraction protocol[23]. Genomic DNA quantity and integrity were assessed using the 1x dsDNA High Sensitivity Assay on a Qubit Fluorometer (Invitrogen), and Genomic DNA ScreenTape on an Agilent TapeStation 4150, respectively, prior to proceeding with genomic library preparation, parasite enrichment and sequencing.

### Multiplexed whole-genome capture and sequencing
We adopted and optimized a multiplexed hybrid capture assay (Supplementary Fig. 2) to selectively enrich whole *P. falciparum* genomes from dried-blood spots prior to deep sequencing according to previously published methods[24]. Custom GC-balanced, biotinylated DNA probes were designed in silico to tile 99.8% *P. falciparum* 3D7 v3.1 reference genome using the Roche NimbleGen SeqCap EZ Designs v4.0 (Madison, USA). To remove probes that hybridized to the human or mosquito vector, they were screened against hg19 and AfunF1 (downloaded from VectorBase). Genomic library preparation, hybridization capture, and sequencing were conducted at the Yale Center for Genomic Analysis (YCGA). Briefly, library preparation for each sample was conducted using a modified Roche/Nimblegen SeqCap EZ Library Short Read protocol[25]. Library concentration was determined using PicoGreen assay (Invitrogen) and size selection was

performed on a Caliper LabChip GX instrument (PerkinElmer). Equimolar amounts of each dual-indexed genomic library were pooled in 4-plex prior to capture for a total of 1 µg total genomic DNA per hybridization reaction. Samples were heat-denatured and mixed with the custom DNA probes (Roche/NimbleGen SeqCap EZ) and hybridization performed at 47 °C for 16 h. Samples were washed to remove non-specifically bound DNA fragments. The captured libraries were PCR amplified and purified with AMPure XP beads. Samples were sequenced using 101 bp paired-end read sequencing on an Illumina NovaSeq 6000 at YCGA with a target of 30 million reads per sample, for an expected *P. falciparum* mean genome coverage of 100X. We used univariate logistic regression to detect correlates of *P. falciparum* capture efficiency and genome coverage.

### Additional genomic datasets

To contextualize Zambian *P. falciparum* genomic diversity within Africa, we included and analyzed 781 publicly available *P. falciparum* WGS data from the Pf3k database from 5 countries (Democratic Republic of the Congo, Ghana, Guinea, Malawi, and Tanzania). Raw Fastq files were downloaded from SRA using pysradb[26] and processed in the same way as the newly sequenced WGS from Zambia. 760 out of 781 genomes were retained after filtering by genome coverage (> 50% of *P. falciparum* genome covered at > 5X read depth). SRR accession numbers are provided in Supplementary Data 7 and Supplementary Data 8.

### Read mapping and SNP discovery

As described by Carpi and colleagues[27], Illumina raw paired-end reads were mapped to the *P. falciparum* 3D7 reference genome[28] using BWA-MEM 0.7.17[29]. Aligned reads were marked for duplicates and sorted using Picard Tools v2.20.8. For variant calling only samples with >50% *P. falciparum* 3D7 reference genome with >5X coverage were included, resulting in a total of 241 *P. falciparum* genomes. Variants were called using GATK v4.1.4.1[30] following their recommended best practices. GATK Base Quality Score Recalibration was applied using default parameters and variants from the *P. falciparum* crosses 1.0 release as a set of known sites[31,32]. We used GATK HaplotypeCaller in GVCF mode to call single-sample variants (ploidy 2 and standard-min-confidence-threshold for calling = 30), followed by GenotypeGVCFs to genotype the cohort. Prior to variant filtering, we scored 1,121,403 SNPs with a VQSLOD > 0 across the 241 genomes. Variants removed include those located in telomeric and hypervariable regions[33], SNPs with >20% missingness, and minor allele frequency (MAF) > 0.02, leaving a total of 29,992 high quality biallelic SNPs. Variants were functionally annotated with SnpEff (version 4.3t)[34] for genomic variant annotations and functional effect prediction.

### Multiplicity of infection and parasite genetic diversity

The within-sample F statistic ($F_{WS}$) (Manske et al., 2012) was calculated for each sample using R moimix package version 2.9[35]. The threshold of $F_{WS} > 0.95$ applied to the 29,992 genome-wide SNPs was used to define monoclonal infections, and $F_{WS} < 0.95$ as polygenomic infections. The association between the proportion of polygenomic infections at the individual cluster level with parasite prevalence was assessed using the Spearman method to compute correlation R values and significance P-values.

### Population structure and genetic differentiation

Principal component analysis (PCA) was performed in R using the SNPRelate package version 1.16.1[36] after removing three samples from the Lusaka, Central and Southern Provinces. Further population structure analysis using a Bayesian clustering algorithm[37] in an admixture model implemented in STRUCTURE version 2.3.4 was performed to identify population clusters (K) and genotype clustering according to geographical origin. STRUCTURE was run with a burn-in period of 50,000 followed by 50,000 Markov Chain Monte Carlo iterations. Evanno's method of delta K (ΔK) statistics implemented in the STRUCTURE HARVESTER were then used to determine the most likely genetic clusters. The Cluster Markov Packager Across K (CLUMPAK)

web-based server[38] was used for summation and graphical representation of STRUCTURE results.

### Isolation-by-Distance Analysis Using Mantel Test

Sample's FASTA file was converted from VCF file using Spider. Pairwise genetic differentiation ($F_{ST}$) between populations was calculated using R PopGenome package version 2.7.5[39]. Centroid geographic locations of populations were used for calculating pairwise geographic distance. Mantel Test, i.e., linear regression between pairwise $F_{ST}$ and pairwise geographic distances, was performed to inspect the support for Isolation-by-Distance pattern.

### Parasite relatedness using IBD estimates

Relatedness estimates were based on the expected fraction identity by descent (IBD), a probabilistic measure of the fraction of the genome inherited by a pair of parasites from a recent common ancestor. For all pairwise comparisons of parasite samples across Zambia, we estimated IBD using isoRelate[21], which infers IBD estimates under a probabilistic model that accounts for recombination. MAP and PED files were generated by assuming a constant recombination rate of 13.5 kb per centimorgan (cM) using the moimix package[35], and 27,231 genome-wide SNPs spanning chromosomes 1–14 retained after isoRelate filtering were used as input for downstream IBD analysis. MOI = 1 vs. MOI > 1 status in the PED file was determined using the threshold of $F_{WS} > 0.95$. IBD segments were inferred and reported for genomic regions >50 kb in length, containing >20 SNPs, and with an error rate of 0.001. IBD per SNP was also calculated at the national and provincial levels. Networks of highly related parasites were identified using the igraph package[40]. The pairwise spatial distance (km) between highly-related parasite pairs was measured from the geographic coordinates of sample collection sites at the cluster level using Geographic Distance Matrix Generator Java package[41], and used to visualize parasite IBD-based relatedness across Zambia.

### Detection of selection signatures

We calculated genome-wide test statistics (XiR,s), where XiR,s is the chi-square distributed test statistic for IBD sharing from IsoRelate at SNPs as described by Henden et al.[21] P-values were calculated for XiR,s and –$\log_{10}$ transformed to investigate the significance of selection signatures. We used Gao et al.'s simpleM method[42] to calculate the effective number of independent SNPs across the genome to derive the 5% genome-wide significance threshold. We first calculated composite LD among SNPs from individuals with MOI > 1 to capture the correlation among SNPs, and then derived the $M_{eff}$ using the number of principal components for every 1000 SNPs that capture 99.5% of variation. The simpleM procedure generated a consistent estimation of $M_{eff} = 184$ for every 1000 SNPs, which translates to $M_{eff} = 5010$. Therefore, the 5% genome-wide significance threshold was set to $0.05/ M_{eff} = 10^{-5}$ for scanning positive selection. Regions of high IBD were visualized using Manhattan plots in R and gene annotation was performed using PlasmoDB.

To evaluate the possibility that non-uniform SNP density may contribute to the IBD-based selection signals, we examined the relationship between SNP density patterns and selection signals. SNP-density distributions, calculated within 1 kb vicinity of the focal SNP, were compared between IBD-based non-significant ($P > 10^{-5}$) and significant sites ($P < 10^{-5}$).

To ensure the credibility of IBD selection analyses, we used another algorithm, integrated haplotype score (iHS)[43,44], for inferring selection using the R package, rehh[45]. Since iHS relies on phased data, we restricted the iHS analysis to the 50 monogenomic samples. We define a genomic region as being under selection if it contains at least two extreme markers with iHS values above 5% genomic significance level.

### Calculation of copy number variation (CNV)

Read counts per coding sequence (CDS) of all annotated Pf3D7 genes were calculated using featureCounts[46] and normalized by CDS lengths for

monoclonal samples. The median coverage per sample was used as the reference for copy number = 1. Inferred copy numbers per gene per sample were then obtained by its coverage divided by the median coverage of the sample. Lastly, median, variance, and coefficient of variation of CNV per gene were calculated.

Unless otherwise stated, all references to an analysis in a 'package' indicate the analysis was performed in R version 4.3.0 Where appropriate, outcomes of interest were visualized using the ggplot2 package in R.

### Reporting summary

Further information on research design is available in the Nature Portfolio Reporting Summary linked to this article.

## Results

### Multiplexed *P. falciparum* genome capture and variant discovery

Whole genome capture and sequencing of 459 *P. falciparum* DBS samples collected during the 2018 Zambia Malaria Indicator Survey (MIS) were performed using a 4-plex *P. falciparum* genome capture method (Supplementary Fig. 2). *P. falciparum* parasitemia, estimated by PET-PCR, was highly variable (median = 100 parasites/µL, range: 0.6–143,914 p/uL), and the number of sequenced samples between provinces varied as a function of sampling efforts and parasite prevalence (Supplementary Fig. 1). The median whole genome capture efficiency (the proportion of sequence reads mapping to the *P. falciparum* 3D7 reference genome) was 77.8% (range: 5.4–99.2%) with a mean genome coverage of 53X (range: 0.03–719X) (Supplementary Fig. 3). Parasitemia was a significant predictor of capture efficiency and genome coverage in a univariate quasi-Poisson model (p-value < 0.001) (Supplementary Fig. 4A). Notably, enrichment of the *P. falciparum* whole genome proved inefficient when parasitemia fell below 10 parasite per microliter (Supplementary Fig. 4B). An optimized GATK v4.1.4.1 pipeline[27,30] with some modifications (see Methods for details) was used for variant discovery in samples with at least 50% coverage of the 24 Mb reference genome at a minimum read depth of 5X, resulting in 241 *P. falciparum* WGS, with 238 originating from the well-represented seven out of the ten provinces, and the remaining three provinces in Zambia contributing a single sample each (Supplementary Data 1). Sample missingness (columns) and SNP missingness (rows) were calculated from the VCF file. Supplementary Fig. 5 illustrates the distribution and thresholds (0.2 sample and SNP missingness, and 0.02 minor allele frequency filtering (MAF)) used to identify samples and variants in the data that had a high degree of missingness and were omitted. We retained 29,992 high quality genome-wide biallelic SNPs (Supplementary Fig. 6) distributed across the 14 *P. falciparum* chromosomes (Supplementary Fig. 7) and the apicoplast (not shown) for downstream analyses.

### Rate of polygenomic infections correlates with local parasite prevalence

Our analysis revealed a predominance of polygenomic infections, representing 77% (186/241) of all sequenced samples, suggesting endemic transmission and high levels of superinfection and co-transmission by mosquitoes across the country. While there was some variability at the provincial level (medians ranging from 60 to 87%) (Fig. 1A), we found substantial variation in the rate of polygenomic infections at the sampling cluster level with medians ranging from 20 to 100% (Fig. 1B). The rate of polygenomic infections was, thus, positively correlated with parasite prevalence at the cluster level (Fig. 1C, Supplementary Data 2), but not at the provincial level (Supplementary Fig. 8, Supplementary Data 3).

### Parasite population shows structure at the country and regional levels but not at the provincial level

Principal component analysis (PCA) was conducted on the genome-wide SNPs from the 238 samples to describe genetic clusters. The first two principal components accounted for 28% of the variance (Supplementary Fig. 9). The sample distribution on PCA indicated no clear evidence of geographical clustering of the parasite populations (defined as all samples from a particular province), except for a few outliers from Western and Copperbelt Provinces (Fig. 2A). Model-based population structure analysis implemented in the STRUCTURE program[37] also failed to detect any population structure irrespective of the choice of K (Fig. 2B), with the exception of a sign of genetic admixture in samples from Western Province and to a lesser extent in Copperbelt Province (K = 3).

We further explored population differentiation between parasites collected from the different provinces using $F_{ST}$, a standard measure of divergence between populations. Pairwise $F_{ST}$ estimates of parasite populations at the provincial level showed overall low genetic differentiation ($F_{ST}$, range = 0.008–0.052) (Supplementary Data 4). The lowest genetic differentiation was observed between Luapula and Northern Provinces ($F_{ST}$ = 0.008), two provinces with the highest transmission intensity based on epidemiological data and that border each other, while the highest differentiation was detected between North-Western and Copperbelt Provinces ($F_{ST}$ = 0.052), two neighboring provinces with moderate transmission intensity. Mantel Test analysis revealed no strong correlation between genetic distance and geographic distance (Supplementary Fig. 10).

To contextualize *P. falciparum* genome diversity sampled in Zambia within Africa, we analyzed 760 *P. falciparum* genomes from Pf3k[47] from 3 neighboring countries in Central and East Africa (Democratic Republic of the Congo, Malawi, Tanzania) and 2 countries from West Africa (Ghana and Guinea) (Fig. 3A). PCA conducted on 30,532 biallelic SNPs with MAF > 0.02 from 1,001 *P. falciparum* genomes (241 from Zambia and 760 from 5 African countries) revealed both continental and population specific patterns of genetic variation and differentiation. The first two principal components identified distinct country-level clustering with limited overlap that closely resembled the actual geography (Fig. 3B). As expected, the West African *P. falciparum* populations were distinct from all others and, in East-Central Africa, Zambia was juxtaposed between the Democratic Republic of the Congo (DRC) and Malawi/Tanzania (Fig. 3B). Conducting the analysis excluding the West African countries reveals a distinct clustering pattern by country, forming a continuum in the order of DRC, Zambia, Tanzania, and Malawi (Fig. 3C).

### Evidence of high IBD-based relatedness among parasites at the cluster level

Our identity by descent (IBD) analysis revealed an overall low level of relatedness, with only 3.96% (1145/28,920) of sample pairs of genomes displaying at least one block of IBD shared (minimum 3.7 cM, 20 SNPs). 231 out of 241 genomes shared at least 1 IBD segment with other genomes. Overall, we found only 2% (23/1145) of shared pairs representing relatedness within three generations (i.e., sharing at least 5% IBD, calculated as the proportion of IBD segments over genome length[48]) (Fig. 4A). Assuming an average generation interval of 3 months for *P. falciparum*[49], 2% of shared pairs had a common ancestor less than 1 year ago, reflecting a high degree of transmission and recombination between divergent parasites across Zambia. Additionally, the distribution of pairwise IBD blocks across the genome revealed that most segments were centered around a length of 100Kb with very few at the right tail, demonstrating high IBD (Supplementary Fig. 11 inset). This corresponds to approximately 8 cM in genetic distance and suggests a common ancestor around six generations, equivalent to approximately ~1.5 years (Supplementary Fig. 11).

Relatedness network analysis to investigate clusters of infections sharing >5% (Fig. 4A) of their genome IBD, identified 23 parasite pairs related at the level of second cousins and above. A few clusters of highly related parasites sharing their genome IBD > 50% and >90% were identified, including 3 half siblings (MOI > 1) and 8 clonal lineages (MOI = 1)/pairs that shared one clonal lineage (MOI >= 1), respectively (Fig. 4A). Most of these highly related parasite pairs were identified within the same cluster and province, with only one instance of long-distance clonal sharing between non-neighboring provinces (Luapula and Southern Provinces) (Fig. 4B). These suggest that most transmission occurs locally, with occasional long-distance transmission via potential human migration.

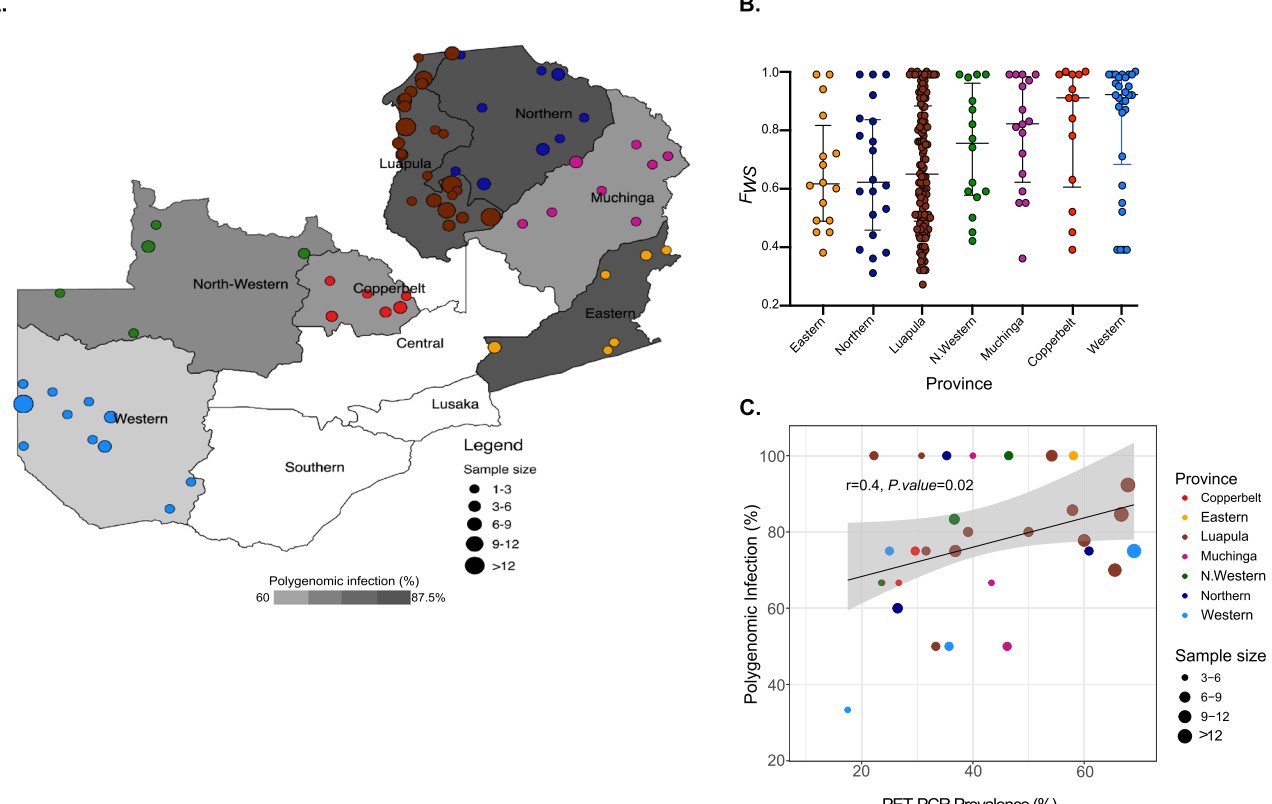

**Fig. 1 | Spatial distribution of 238 *P. falciparum* genomes, polygenomic infections and parasite prevalence across Zambia. A**) Heatmap showing spatial heterogeneity of polygenomic infections in children across Zambia. Heatmap in grey scale illustrates the prevalence of polygenomic infections (range: 60–87.5%). Colored dots represent the sites of sample collections at the cluster level, where circle size is proportional to the number of successfully sequenced samples per cluster. A total of 238 samples were included from 67 clusters across Zambia. **B**) Polygenomic infection estimates based on $F_{WS}$ statistics for each sample by province. The middle lines represent the median value of $F_{WS}$, the whiskers represent the interquartile range, the dots represent individual whole genome sequenced samples and colors correspond to sample geographic origin. **C**) The relationship between PET-PCR *P. falciparum* prevalence and the rate of polygenomic infections at the cluster level. Colored dots denote clusters at the provincial level, with size of the dots reflecting sample size. The grey shaded area around the regression line is the 95% confidence interval. Clusters with less than 3 samples were excluded from the correlation analysis.

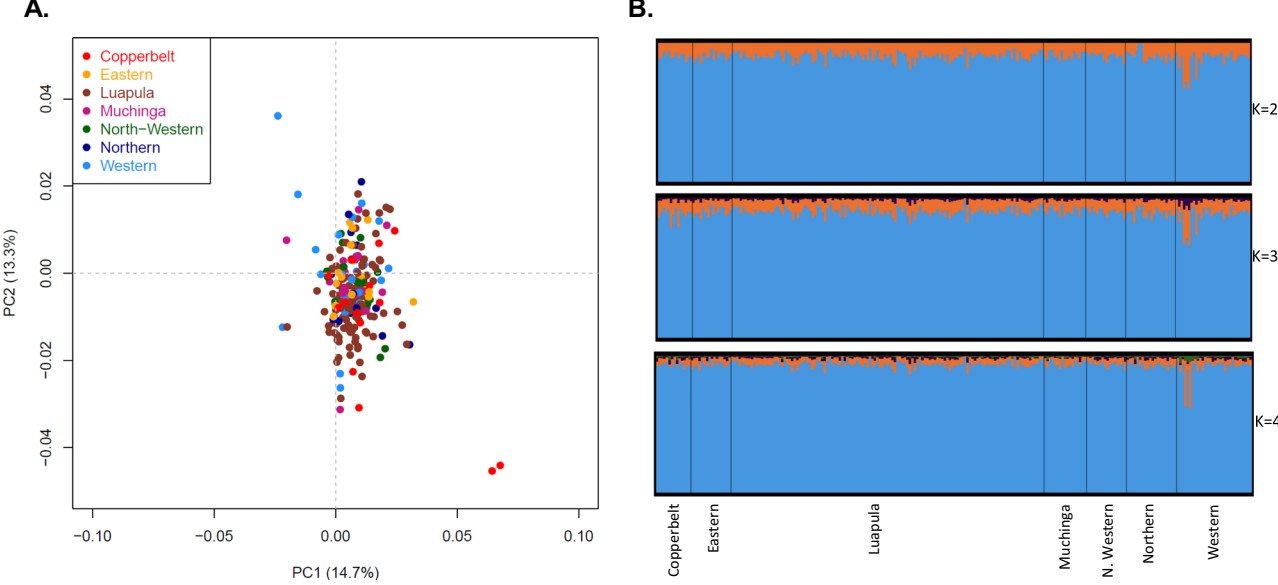

**Fig. 2 | *P. falciparum* population structure and differentiation within Zambia. A**) Principal component analysis of 238 *P. falciparum* parasites across seven provinces in Zambia. Colors indicate geographic origin and dots indicate individual parasites. **B**) Bayesian cluster STRUCTURE analysis of 238 *P. falciparum* parasites. Individual ancestry coefficients are shown for K = 2, 3, and 4, with each vertical bar representing an individual parasite and the membership coefficient (Q) within the province of each parasite population.

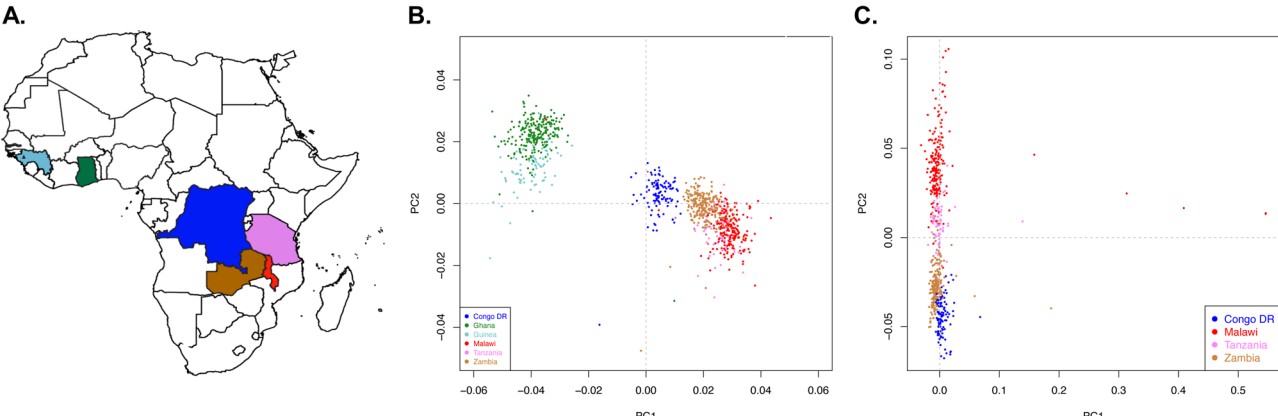

**Fig. 3 | Population structure of *P. falciparum* parasite populations in countries bordering Zambia and in West Africa. A**) Map highlighting parasite populations included in this study. **B**) Separation of West and East African parasite samples using principal component analysis (PCA). The first two principal components generated from 30,532 genome-wide biallelic SNPs across 1,001 *P. falciparum* genomes from 6 countries is shown (explaining 1.3% and 0.9% of the variance in the data set, respectively). Each dot is a sample colored by its geographic origin. **C**) The first two principal components calculated from 30,320 genome-wide biallelic SNPs across 667 *P. falciparum* genomes from Zambia and neighboring countries alone is shown (explaining 0.8% and 0.8% of the variance in the data set, respectively).

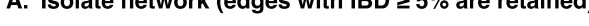

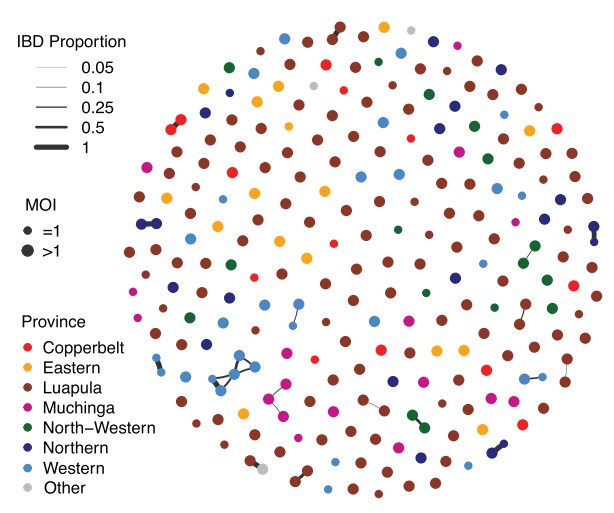

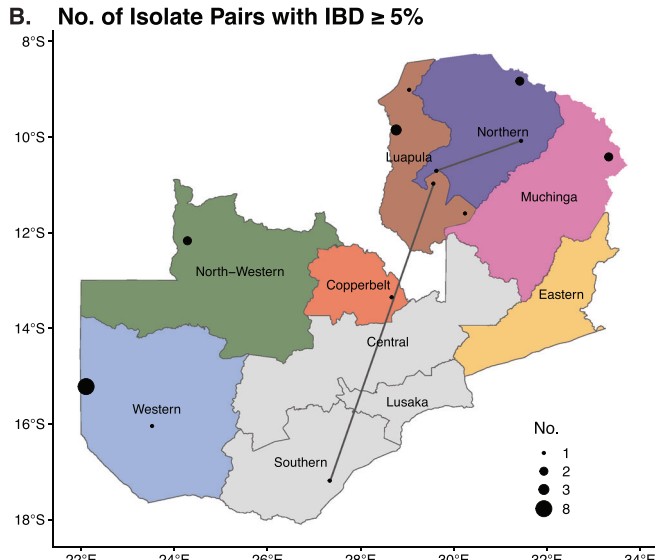

**Fig. 4 | Relatedness networks and patterns of *P. falciparum* parasite connectivity across Zambia. A**) Relatedness network of *P. falciparum* genome pairs having different proportions of Identity byu sharing (5–100%) across Zambia. Each node identifies a unique sample, and an edge is drawn between two samples if their genomes equal or exceed 5% identity by descent (IBD) sharing threshold. Colors indicate the sample geographic origin at the provincial level, which correspond to the same color scheme in **B**. **B**) Spatial distribution and relatedness of parasites that share IBD ≥ 5% at the cluster level. Sample pairs with IBD ≥ 5% within the same cluster are indicated by a black circle, radius of which represents the number of such pairs. Connecting lines between clusters indicate long-distance sharing of IBD.

## Identification of potential selection signals on chromosomes 3, 6, 8, 10 and 12

The genome-wide distributions of pairwise IBD can identify genomic regions that are conserved over time and space and may be indicative of positive selection. We calculated the chi-square distributed test statistic for IBD sharing ($X_{iR}$) at each SNP and plotted the $-\log_{10}$ transformed p-value of these statistics across the genome. Using 5% genome-wide significance threshold (p-value < $10^{-5}$; see Methods for calculation), we discovered 258 significant SNPs and 83 genes with signals of positive selection across six chromosomes (chromosomes 3, 4, 6, 8, 10, 12) (Fig. 5A, Supplementary Data 5). We then identified significantly selected regions by a sliding-window search of 50 kb ranges that contained at least two significant SNPs (Supplementary Data 6). The selected regions were recovered on chromosomes 3, 6, 8, 10 and 12. The overall selection pattern in Zambia resembles the positive selection signature from IBD analyses of Malawi genomes

(Figure 6 in Henden et al.[21]) as well as pyrimethamine-associated selection signal from genome-wide association studies in Senegal (Fig. 3 in Park et al.[50]). Notably, the observed selection pattern in Zambia lacks a commonly selected region on chromosome 7 that encompasses the *pfcrt* gene which occurs in parasite populations from African and Southeast Asian countries. Zambia transitioned from chloroquine to ACT as the first-line drug in 2002[51]. With the current genomic samples from 2018, there has been a continuous 16-year period of drastic reduction in chloroquine usage, resulting in an absence of selection signatures in this region.

Genes with the highest number of significant sites include surface proteins/antigens: *pfclag3.2* (PF3D7_0302200; Chr 3; 29 Significant SNPs), *pfdblmsp2* (PF3D7_1036300; Chr10; 23 SNPs), and *pflsa1* (PF3D7_1036400; Chr 10; 7 SNPs); serine/threonine kinases: *pffikk10.2* (PF3D7_1039000; Chr 10; 11 SNPs), *pfsrpk1* (PF3D7_0302100; Chr3; 8 SNPs); and other conserved proteins: *pf11-1* (PF3D7_1038400; Chr 10; 17

**Fig. 5 | Signature of positive selection across *P. falciparum* genomes in Zambia.** Signature of positive selection across all provinces (**A**), across Eastern Provinces (**B**) and Western Provinces (**C**). Each dot represents a SNP and the colors identify each chromosome. Dashed horizontal lines represent a 5% genomic significance threshold ($p < 10^{-5}$ (i.e., -$\log_{10}$(p) >5)). The selected genes discussed in the results are indicated to the right of each region, except for *pfdhdr* and *pfdhps* (shown in grey), which are proximate to the selected region/locus (**A**). *pfclag3.2* = cytoadherence linked asexual protein 3.2, *pfdhfr* = dihydrofolate reductase, *pfpk4* = eukaryotic translation initiation factor 2-alpha kinase, *pfdhps* = hydroxymethyldihydropterin pyrophosphokinase-dihydropteroate synthase, *pfdblmsp2* = duffy binding-like merozoite surface protein 2, *pflsa1* = liver stage protein 1, *pf11-1* = gametocyte-specific protein, *pfgch1* = GTP cyclohydrolase I gene.

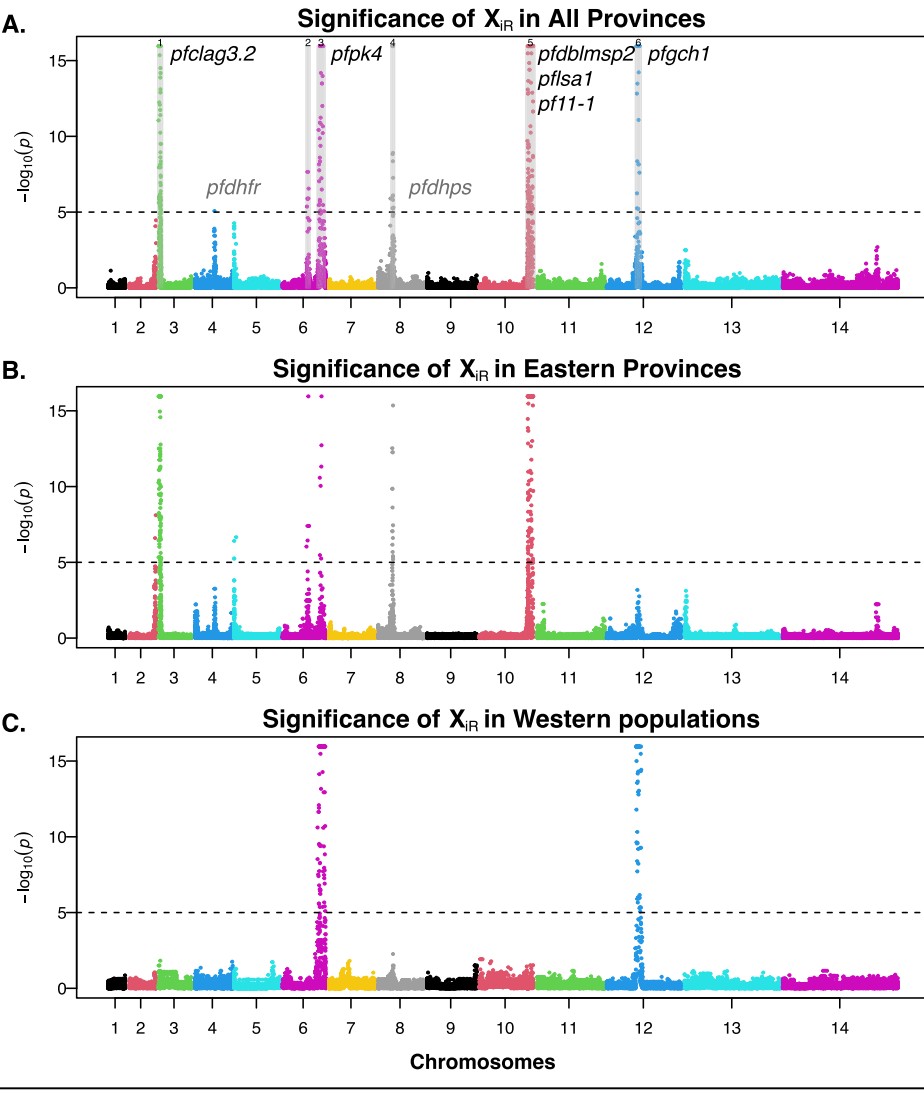

SNPs), PF3D7_0809600 (Chr 8; 10 SNPs), and *pfhect1* (PF3D7_0628100; Chr 6; 8 SNPs).

The selected region in chromosome 3 was marginally significant in Ghana and Malawi, but very robust in our Zambia samples (Fig. 5A), particularly in eastern provinces (Fig. 5B, C). In the isolate relatedness network of this genomic region, a prominent cluster and a smaller cluster of related isolates exist (Supplementary Fig. 12). *pfclag3.1* and *pfclag3.2*, located in this genomic region, play a critical role in erythrocyte invasion during the asexual cycle by regulating solute transport (ions, nutrients, and antimalarial toxins) at the infected erythrocyte membrane[52]. In addition to potential drug resistance properties[53,54], this gene family experiences balancing selection, with rapid evolution via gene conversion between *pfclag3.1* and *pfclag3.2*[52].

Chromosome 4 showcases a lone significant site and a marginally selected region akin to Guinea, Gambia, and Southeast Asia[21]. This region is proximate to *pfdhfr*, which is linked to pyrimethamine resistance. Similarly, the selected region on chromosome 8 is 15Kb upstream of *pfdhps*, which responds to sulfadoxine usage. More than 90% of the Zambian samples have *pfdhfr* (N51I, C59R) and *pfdhps* (A437G) resistant genotypes, indicating widespread and persistent sulfadoxine-pyrimethamine drug use in Zambia.

The chromosome 6 dynamics are marked by two distinct selected regions. The first one, spanning 730kb-840kb, comprises conserved proteins. The relatedness network forms a tight cluster composed of samples from eastern provinces (Supplementary Fig. 12). The second selected region, ranging from 1,040,000 to 1,260,000, is recognized as a long haplotype subject to selection in multiple studies[21,52,55]. *pfpk4* in this region exhibits

significance at four sites. The phosphorylation of eIF2alpha by *pfpk4*, triggered by artemisinin treatment, leads to parasite latency, potentially contributing to the maintenance of the extended haplotype[56]. Within this region lies the gene *pfaat1* (PF3D7_0629500), which bears the S258L mutation that segregates at medium frequency. Despite the gene not having a significant selection signal, S258L is associated with chloroquine resistance[57] and the gene plays a crucial role in the efflux of drugs[58].

The region on chromosome 10 potentially reflects the influence of *pfmspdbl2*, encoding a merozoite surface protein containing a Duffy binding-like (DBL) domain. Overexpression of *pfdblmsp2* imparts resistance to halofantrine and cross-resistance to mefloquine and lumefantrine[59,60]. As mefloquine and lumefantrine can be the long-lived paring drug in artemisinin-based combination therapy, the copy number variation of *pfdblmsp2* potentially undergoes selection in response to persistent use of ACT[59]. Other selected genes include *pf11-1*, critical for gametogenesis[61], and *pflsa-1*, a liver-stage antigen, as evidenced by positive selection from the McDonald-Kreitman test[62].

On chromosome 12, the selection signals are likely associated with the sustained utilization of the sulfadoxine-pyrimethamine as the front-line antimalarial drug for intermittent preventive treatment of malaria in pregnancy (IPTp). Copy number variation in *pfgch1* has been found to confer pyrimethamine resistance[63] and compensate for the cost of drug-resistant mutations in the less efficient dihydrofolate reductase (*dhfr*) and dihydropteroate synthase (*dhps*) enzymes[64]. Notably, strong signals are observed in uncharacterized genes PF3D7_1223400 and PF3D7_1223500,

aligning with findings from a selection study focused on prolonged sulfadoxine-pyrimethamine usage in Malawian parasites[65]. The full list of genomic regions under positive selection is provided in Supplementary Data 5 and Fig. 5A. In addition, there was some variation in genomic regions under selection between eastern and western provinces, which constitute two transmission zones in Zambia (Fig. 5B, C, Supplementary Fig. 12), suggesting that parasites may experience different selection pressures due to exposure to different control interventions, mosquito vectors, and environmental conditions.

We extended our investigation into selection using the standardized integrated haplotype score (iHS)[43,44], and identified 11 genomic regions exceeding the 5% genomic significance threshold (Supplementary Fig. 13). Comparative analysis of selected SNPs and regions using XiR and iHS revealed somewhat similar selection patterns (Supplementary Fig. 13A, B), with shared selection on chromosomes 6 and 8. Discrepancies in the identified selected SNPs and regions stem from the methods' distinct detection capabilities: iHS excels at detecting older selection, given its extensive surface antigen gene list, while $X_{iR}$ (isoRelate) focuses on detecting selection within the last 200 generations, corresponding to the recent 60 years of parasite evolution[21,66].

## Discussion

Robust routine epidemiological and genomic surveillance is essential to successful malaria control and elimination efforts[3]. While unlikely to be implemented routinely in sub-Saharan Africa, *P. falciparum* WGS provides the richest possible data on parasite populations for quantifying measures of mixed infections, parasite population differentiation, spatial mixing, selection, and other similar metrics not available using less granular and targeted genomic approaches. Here, we describe the largest collection of *P. falciparum* genomic sequence data collected during the 2018 national MIS from ten provinces across Zambia.

The rate of mixed infections is relevant for understanding regional malaria epidemiology. Mixed infections, also known as multiplicity of infections (MOI), are indicative of intense local exposure rates and correlate with estimates of malaria prevalence within Africa[67,68] and can range from one (monogenomic infection) in low transmission settings to MOI > 10 (polygenomic infection) in high transmission settings[69]. Comprising 77% of all sequenced samples, polygenomic infections ($Fws < 0.95$) dominated across Zambia, suggesting that malaria transmission remains high across the country with superinfections and co-transmission also likely to be high, even though malaria incidence has decreased since 2011[70]. Although there was limited heterogeneity of polygenomic infection rates at the provincial level (Fig. 1B), we found a positive correlation between the prevalence of polygenomic infections and parasite prevalence at the sampling cluster level (Fig. 1C). which agrees with a previous study[68], and especially in regions where malaria transmission is highly heterogeneous. Thus, MOI derived from WGS is an appropriate indicator to evaluate the success of malaria control activities since any control measures that reduce parasite prevalence will reduce the likelihood of mosquitoes taking multiple infective feeds such that control efforts are expected to reduce MOI and ultimately within-host parasite diversity[71].

Using classical genetic metrics (Wright's fixation index ($F_{ST}$) and STRUCTURE), we identified high population level diversity across seven provinces consistent with a panmictic population, i.e., parasites are not clustered based on their geographic origins, suggesting parasite migration and gene flow between and within provinces across Zambia despite the marked reductions in malaria incidence recorded over the last decade and the highly heterogeneous transmission across provinces[72]. This is not unexpected considering that $F_{ST}$ has been shown to be less reliable in detecting small-scale population structure in malaria compared to other metrics[11]. Nevertheless, this suggests that parasite diversity in these seven provinces has not been strongly influenced by current control measures and that without further significant transmission reduction measures aimed at fragmenting parasite populations, subnational elimination will be challenging. This is similar to other studies where parasite genetic diversity did not

strongly correlate with local transmission intensity[73,74]. Considering the limited range that African malaria vectors routinely disperse (a maximum of 10 km)[75], it is likely that human movement plays a major role in maintaining a diverse gene pool with low genetic differentiation. Different environmental variables (geographic distance and other landscape parameters) and human movement patterns may affect parasite migration and gene flow among different geographic areas[76]. One of the limitations of our study is that travel histories from malaria cases were not collected so the directionality of parasite spread could not be determined. Nevertheless, we can assume limited travel associated with our study subjects as they were children younger than 5 years of age. An additional limitation is the low number of malaria positive DBS samples that were obtained from the Southern, Central and Lusaka Provinces, provinces with the lowest malaria burden, which affected the numbers of samples that could be sequenced.

After identifying a panmictic Zambian population, we investigated the continental population structure and found distinct geographical clustering (Fig. 3B) that essentially mirrored the physical geography, placing Zambia in proximity to its neighbors and isolated from more distant West African parasite populations. This finding reinforces the need for cross-border coordination to maximize the impact of malaria control and elimination efforts. Despite the two countries sharing a border, parasites from Malawi and Zambia clustered separately in the PCA plot (Fig. 3C), which suggests low parasite migration and gene flow patterns between these countries. However, factors such as variation in utilization of control measures, and year of sample collection (i.e., the observed structure may be due to temporal rather than spatial differences as samples from these two countries were collected at different times) could contribute to this observed population structure between Malawi and Zambia.

Notably, although the PCA did not reveal geographic clustering of parasite populations within Zambia, the IBD-based relatedness measures provide a more local-scale of isolation by distance, as IBD and SNP PCA are measures of different evolutionary times. Unlike classical population genetic metrics, IBD-based relatedness measures recent recombination events (within 12 generations) and genomic signal changes due to recent selection pressures (within 200 generations). Using a hidden Markov model (HMM) algorithm implemented in the isoRelate software in R package[21], most Zambian parasite pairs had low relatedness (sharing 0–5% of their genome by IBD), which implies parasites originating from two unrelated oocysts[71] and evidence of high recombination between divergent parasites - findings to be expected in high transmission settings[15]. However, 23 parasite pairs exhibited relationships at the second cousin level and beyond. We identified several clusters of highly related parasites (genomes with IBD values exceeding 50% and 90%, equivalent to half siblings and clonal lineages), suggesting some level of inbreeding or local transmission at the cluster level in some provinces[77]. This result is in agreement with other study findings where IBD-relatedness estimates correlated with inter-clinic distance and detected spatial patterns of malaria parasite connectivity at a small spatial scale[11]. Interestingly, we identified one instance of long-distance clonal sharing between distant non-neighboring provinces, Luapula and Southern Provinces, suggesting that while most transmission occurs locally, some occasional long-distance transmission via potential human migration can be detected.

Malaria control measures exert strong evolutionary selection pressures on parasite populations that can be identified by IBD analysis[78]. Hence, the detection of loci under directional selection (selective sweep)[79] from WGS data is one approach to identify such selection signals in known and new drug resistance mutations, vaccine candidate antigens, and other genes that impact life cycles[80,81]. Significant selection regions were identified on chromosomes 3, 6, 8, 10, and 12 in the Zambian parasites. To address potential bias from non-uniform SNP density across the genome, we analyzed the relationship between SNP density and IBD selection signals. The overlap in SNP density distributions for significant and non-significant sites indicates a conservative estimate (Supplementary Fig. 14A). The higher proportion of SNP-dense regions in chromosomes 3 and 10 are attributed to one gene within the region,

where the median SNP densities of the regions are not high (Supplementary Fig. 14B). PF3D7_0302300 is a pseudogene, whereas *pfdblmsp2* is indeed an important gene for drug resistance to mefloquine and lumefantrine through functional experiments[59,60]. Excluding these regions from the IBD analyses still yields significant results (Supplementary Fig. 14). These results indicate the robustness of detected selection.

The selection patterns in Zambia lacked a commonly selected region on chromosome 7 (*pfcrt*), contrasting with parasite populations from other regions. Similarly, we did not observe selection signature in *pfaat1*, the second important transporter gene for chloroquine resistance[57]. This absence is attributed to the country's transition from chloroquine to ACT 16 years ago, signifying a shift to chloroquine-sensitive *P. falciparum* parasites. Indeed, we found strong selection signatures as well as copy number variation (CNV) in two genes, *pfpk4* and *pfdblmsp2*, which confer resistance to artemisinin or its pairing drug (i.e., lumefantrine) (Supplementary Data 9). The strongest genome-level selection signature comes from resistance to sulfadoxine-pyrimethamine (SP). In addition to the marginally selected region on chromosome 4 proximate to *pfdhfr* and the selected region on chromosome 8 near *pfdhps*, *pfclag3.1*, *pfclag3.2* on chromosome 3 and *pfgch1* on chromosome 12 also indicate selection on resistance to SP. Interestingly, CNV is also present in *pfclag3.1 and pfclag3.2* but not in *pfgch1* (Supplementary Data 9). The presence of selection signals for SP sites suggests that Zambian parasites are under strong selection from sulfadoxine-pyrimethamine usage for IPTp. This finding warrants close monitoring of the emergence and spread of SP and active surveillance of drug resistance for artemisinin and its pairing drug in Zambia.

## Concluding remarks

Using a multiplexed whole genome capture and sequencing approach, we generated the largest collection of whole genome data from *P. falciparum* infections across Zambia. The parasite genomic signals from this study, such as high polygenomic infections, low IBD-based parasite relatedness, and lack of population structure across Zambia despite clear epidemiological zones, reflects regional and local levels of endemicity and ongoing transmission intensity. Our findings support malaria parasite genomic metrics commonly reported in African *P. falciparum* parasite populations (*i.e.*, high genetic diversity and MOI, low IBD relatedness, and parasite outcrossing). Importantly, we detected a continuum of parasite population differentiation between East and Central Africa, suggesting that standing genetic variation and selection may contribute to the observed geographic-specific patterns of genetic differentiation, which in turn can be harnessed to infer the origin of parasites at the country level. As malaria control efforts intensify and persist, we anticipate the emergence of highly fragmented parasite populations at provincial, district, or village levels. This fragmentation increases the feasibility of achieving subnational malaria elimination in Zambia. Moreover, the identification of putative signals of positive selection in several genes, including antimalarial drug resistance genes, warrants continued surveillance. Overall, this study demonstrates the utility of whole-genome sequencing of nationally representative *P. falciparum* infections and population genomic analyses to provide insights into malaria transmission dynamics at different spatial levels and improve our understanding of how parasites evolve in the face of interventions.

## Data availability

The newly generated sequence data are available in the NCBI Sequence Read Archive under BioProject PRJNA932927. Source data for the figures are available in Supplementary Data 1–9 and from https://doi.org/10.5281/zenodo.10891196[82].

## Code availability

Key analysis scripts can be accessed at https://doi.org/10.5281/zenodo.10891196[82] along with intermediate files.

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

## Acknowledgements

The authors are grateful to the Zambian communities, particularly the volunteers and their families, for providing samples during the MIS. We would like to thank the staff of the Zambia National Malaria Elimination Centre for their ongoing support, especially the field researchers who conducted the nationwide survey. The authors thank Irina Tikhonova, Christopher Castaldi and Kaya Bilguvar of the Yale Center for Genomic Analysis for technical support on the optimization of the multiplexed hybrid capture of *P. falciparum* genomes from DBS samples. We would also like to extend our gratitude to the communities and researchers of malaria endemic countries that enabled the collection and availability of the *P. falciparum* genomes used in this study made publicly available through the MalariaGEN *P. falciparum* Community Project. This work was supported by funds to G.C. from the Purdue Department of Biological Sciences. D.J.B. discloses support from the Bill & Melinda Gates Foundation through a grant to PATH (OPP1134518 / INV-009984). The Southern and Central Africa International Center of Excellence for Malaria Research (W.J.M.) was supported by funding from the National Institute of Allergy and Infectious Diseases (U19AI089680).

## Author contributions

G.C. and D.J.B. contributed to funding acquisition, project resources and supervision. G.C., W.J.M. and D.J.B., conceived and designed the study. A.A.F., D.J.B., D.E.N., W.J.M. and G.C., coordinated sample selection and curation. M.C.M., B.M., C.M. M.H. and D.J.B. collected samples and epidemiological data. A.A.F., J.D. and I.C. performed laboratory analysis. S.X., K.P.B., J.T., Q.H. and G.C. performed and supervised bioinformatics analysis. Q.H., A.A.F. and G.C. contributed to formal genomic analysis, visualization, interpretation and writing the original draft. All authors contributed to review and editing.

## Competing interests

The authors declare no competing interests.
