## [Peer Review File · Communications Medicine]

Reviewers' comments:

Reviewer #1 (Remarks to the Author):

This paper describes a genome-wide survey of *P. falciparum* parasites in Zambia and the results of multiple analyses of that dataset. The dataset is an important one, the analyses are appropriate and seem well done, and the paper is well-written. While most of the results do not suggest major discoveries, it is an important study for understanding malaria in Zambia and for providing a large dataset for clarifying the relationship between genetics and transmission.

I only have one substantial issue to raise about the study, and that concerns the identification of signals of selection. The method used to search for selection is generally sound and most of the highly significant findings are likely real, since they represent known or likely sweeps. However, there are highly significant novel signals identified as well, in particular the novel loci on chromosomes 3 and 10. My concern is that this kind of IBD test can be misleading when SNP density is nonuniform: the algorithm can identify very short IBD segments in areas with very high SNP density, segments that would not be detected elsewhere and that might not be meaningful. Based on the plot of SNP density, both of these signals do seem to be in regions of high SNP density, although how high is not clear. To eliminate this possibility, then, I would strongly suggest that the authors look at the fine-scale distribution of SNPs in these regions. If the density is very high compared to the genome average, the test should be repeated after thinning the SNPs in order to rule out an artifactual source of signal.

Minor (mostly very minor) issues:

The sampling framework should be explained in the text – in particular, what sample clusters are.

l. 60: The end of the sentence ('as a result of') does not really explain what immediately precedes it (lack of population structure).

l. 124: 'high-level' should be 'a high level'.

l. 126-8: The text does not correspond to the figure panels: (A) is a map of sample sizes, not the rate of polygenomic infections.

l. 131: It's not clear what 'relatively high' genetic diversity means here or why the observation supports that conclusion. Also, some comment about the prominent bands visible in Fig. S10 would be helpful. Do they represent clusters of relatives?

l. 172: The sentence would be better without 'distance' in it.

l. 183: Should be 'Mean IBD between clusters', yes?

l. 246: DBS has not been defined yet.

Fig. S3. Whiskers are said to represent the range but in some cases clearly do not represent the full range.

Fig. S6 legend: SNP and sample missingness should be defined somewhere. Also there is a typo in the fourth line ('rom').

Fig. S8 legend title: Figure does not contain information about allele frequencies.

Fig. S14 legend is confusing: it talks about cM, but the actual plot is in kb and the implications of genetic distance don't seem to be stated anywhere.

Reviewer #2 (Remarks to the Author):

In this manuscript, Fola and colleagues describe the genomic diversity and relatedness of *Plasmodium falciparum* parasites sampled across Zambia. The approach is not novel, but the data set is new and provides previously unavailable insight into parasite dynamics in Zambia. Genomic studies of this vein establish critical baseline understandings that can be used later to assess the efficacy of disease interventions. I offer a few points to make the analyses more robust, but overall, I found the approach thorough and appropriate.

IBD analysis:

- While isoRelate can accommodate multiclonal infections with a diploid model, its power depends on the number of genomes and the genome ratios. As a check to ensure the data are within reasonable bounds, the authors could compare the IBD patterns obtained with monoclonal-only vs polyclonal samples.

- Precise language isn't consistently used. For instance: "only 3.21% (906/28,203) of pairs of genomes displaying at least one block of IBD shared". Technically this is pairs of *samples*, many of which contain multiple genomes. This may appear to be splitting hairs, however, such differences will be important later if this study is used as a baseline for understanding future epidemiological patterns.

Selection analysis:

- As with the IBD analysis, I would advise more exact language in regards to the selection analysis. These are all *putative* or *potential* signatures of selection. At times this more circumspect language is dropped.

- It is worth noting that the isoRelate authors suggests coupling its selection test with other selection tests to obtain the most robust results,

- Can anything more be said about the siblings and clones? This struck me as one result that might have implications for understanding local transmission and could potentially be explored further.

- The authors posit that there may be different selection pressures across the country. I think they could additionally explore other hypotheses like local patterns of relatedness and differences in power.

- To my eyes, the signals picked up by the XiR approach agree with previous selection scans in Africa. Are there any loci that are novel? Are there any loci that are likely artifacts?

PCA:

- The PCA would be more robust if highly related samples (ie, clonal or nearly clonal) are pruned. Otherwise, these pairs can artificially drive the partitioning. This doesn't appear to be an issue with the results, but it would still be best practices to do this.

- How is polyclonality handled in the PCA?

Other small edits:

- It could help to define "cluster" early in the manuscript. Since this isn't a term I usually associate with sampling location, I was initially confused.

- On page 8 (top), the text reads "Mean IBD within clusters showed a strong negative correlation with inter-cluster geographic distance (Figure 4D)". I assume this should be: "Mean IBD *between* clusters..."

Reviewer #3 (Remarks to the Author):

This paper is very well written and it highlights novelty in the area of parasite genomics. The paper highlights some interesting findings.

The methods section is clearly written and concise, and the results are also very well written though overall, it would have been nice for the authors to highlight further on the drug resistance markers that were identified as these seem to key in the fight against malaria and they would also be very good for control recommendations.

November 28, 2023

COMMSMED-23-0188-T

Dear Editor and Reviewers,

Please find enclosed the revisions we made to our manuscript entitled “**Genomics reveals heterogeneous *Plasmodium falciparum* transmission and population differentiation in Zambia and bordering countries.**” We are grateful to the constructive suggestions and thoughtful comments made by each of the reviewers and the Editor.

Our revised manuscript provides additional clarity, specifically regarding the sampling framework and methodology, as well as a revised IBD analyses and supplementary analyses to corroborate our findings on the signatures of selection in the Zambian parasites and contextualize our findings with existing literature. Furthermore, to better address polygenomic infections, we have conducted supplementary analyses. Specifically, we have undertaken PCA and IBD analyses separately for monoclonal and polygenomic infections, which overall support our findings from the analysis conducted on the full dataset. These additional efforts aim to strengthen the robustness of our conclusions. Finally, we have added the link to the github repository that contains the codes used for the analyses included in the revised manuscript: https://github.com/giocarpi/Pf_wgs_Zambia.

We provide detailed responses to the reviewers’ comments below. The line numbers in our point-by-point responses reflect the line numbers in the submitted revised manuscript. We provided in addition to the revised and clean manuscript, a version with all track changes highlighted from the original version of the manuscript to facilitate the reviewers given the extensive revisions.

Reviewer #1 (Remarks to the Author):

*This paper describes a genome-wide survey of *P. falciparum* parasites in Zambia and the results of multiple analyses of that dataset. The dataset is an important one, the analyses are appropriate and seem well done, and the paper is well-written. While most of the results do not suggest major discoveries, it is an important study for understanding malaria in Zambia and for providing a large dataset for clarifying the relationship between genetics and transmission.*

I only have one substantial issue to raise about the study, and that concerns the identification of signals of selection. The method used to search for selection is generally sound and most of the highly significant findings are likely real, since they represent known or likely sweeps. However, there are highly significant novel signals identified as well, in particular the novel loci on chromosomes 3 and 10. My concern is that this kind of IBD test can be misleading when SNP density is nonuniform: the algorithm can identify very short IBD segments in areas with very high SNP density, segments that would not be detected elsewhere and that might not be meaningful. Based on the plot of SNP density, both of these signals do seem to be in regions of

high SNP density, although how high is not clear. To eliminate this possibility, then, I would strongly suggest that the authors look at the fine-scale distribution of SNPs in these regions. If the density is very high compared to the genome average, the test should be repeated after thinning the SNPs in order to rule out an artifactual source of signal.

Response: We appreciate the reviewer's insightful comments and suggestions that helped us improve our manuscript. The reviewer makes an insightful observation about the SNP density and the IBD-based approach to investigating evidence of selection. As suggested, we carefully examined the relationship between SNP density and selection signals in our data and found that the SNP density distributions of significant vs. non-significant sites largely overlap with each other (Fig R1A). The higher proportion of SNP-dense regions in chromosomes 3 and 10 are attributed to one gene within the region, where the median SNP densities of the regions are not high (Fig R1B). PF3D7_0302300 is a pseudogene, whereas *pfdblmsp2* is indeed an important gene for drug resistance to mefloquine and lumefantrine through functional experiments (see the new results section "Identification of potential selection signals on chromosomes 3, 6, 8, 10 and 12"). We also added comparisons to other studies in this section: the selected regions on chromosomes 3 and 10 were also inferred in other studies: Ghana, Malawi and Laos have the selection signature on chromosome 3 regions and Malawi, Mali, DRC have the selection signature on chromosome 10 regions respectively in the original isoRelate paper. The chromosome 10 region containing *pfdblmsp2* also had strong association with halofantrine, mefloquine and lumefantrine drug usage in GWAS studies (Van Tyne et al. 2011). Given that the minimum length of IBD segment in our analyses is 50 kb (so that short IBD segments are excluded to lower FDR), we do not think the SNP density impacted the credibility of the IBD inference.

Fig R1. SNP density patterns of different significance status, chromosomes, and genes. (A) Comparison of SNP-density distributions, calculated within 1 kb vicinity of the focal SNP, between

non-significant ($P > 10^{-5}$) and significant sites ($P < 10^{-5}$). (B) Distributions of SNP-density on different chromosomes.

To ensure that the selection signal was not due to high SNP density, we performed IBD analyses again with regions of PF3D7_0302300 and *pfdblmsp2* excluded. We found that regions on chromosomes 3 and 10 are still significant with a smaller number of significant sites (Fig. R2). Therefore, we believe that the result presented in Figure 5 is a conservative estimate of regions under selection.

Fig R2. Selection signal of 241 samples with regions of PF3D7_0302300 and *pfdblmsp2* excluded.

Minor (mostly very minor) issues:

The sampling framework should be explained in the text – in particular, what sample clusters are.

Response: Thank you for your suggestion. We added additional information regarding the sampling framework in the Methods section, and specifically clarified sample clusters (P. 17, L. 411-418).

l. 60: The end of the sentence ('as a result of') does not really explain what immediately precedes it (lack of population structure).

Response: We have sought to clarify the text in this section which now states “High transmission intensity is associated with high levels of MOI, high genetic diversity, low parasite relatedness, and limited population structure due to extensive parasite recombination rates⁸” (P. 3. L. 62-64).

I. 124: 'high-level' should be 'a high level'.

Response: We corrected this and other typographical errors in the revised manuscript.

I. 126-8: The text does not correspond to the figure panels: (A) is a map of sample sizes, not the rate of polygenomic infections.

Response: Thank you for your suggestion. In Figure 1, panel A shows both sample size (size of colored dots) and the rate of polyclonal infections (heatmap in grey scale). We modified the legend of Figure 1 to better clarify the illustration of these two metrics in panel A.

I. 131: It's not clear what 'relatively high' genetic diversity means here or why the observation supports that conclusion. Also, some comment about the prominent bands visible in Fig. S10 would be helpful. Do they represent clusters of relatives?

Response: We thank the reviewer for noting this detail. The genetic distance here refers to the percentage of differences among SNPs only, thus representing relative divergence among samples instead of pi of whole genomes. Given this information is also represented in PCA and Fst analyses, we removed this analysis and the related figure.

I. 172: The sentence would be better without 'distance' in it.

Response: We have removed the word ‘distance’ as suggested. This was a typographical error. We corrected this in the revised manuscript as : “Assuming an average generation interval of 3 months for *P. falciparum*²⁷, 2% of shared pairs have a common ancestor less than 1 year ago, reflecting a high degree of transmission and recombination between divergent parasites across Zambia (P. 8, L. 185-188).

I. 183: Should be 'Mean IBD between clusters', yes?

Response: Yes, we meant “Mean IBD between clusters”. However, we found an error in the original pairwise IBD vs. geographic distance analyses and redid the analyses. We found IBD above 5% only at the local cluster level with only two examples of long-distance sharing: one within the province and one between two provinces. The results show that high IBD sharing is almost exclusively limited to very short distances. We therefore removed the original Fig 4D.

I. 246: DBS has not been defined yet.

Response: We added the abbreviation for dried blood spot (DBS) in the Introduction (P. 4, L. 93-94), which precede the occurrence of this abbreviation.

Fig. S3. Whiskers are said to represent the range but, in some cases, clearly do not represent the full range.

Response: Thank you for this observation. The whiskers represent the range excluding outlier and updated the Figure S3 legend accordingly. We also specified that five samples (2 samples out of 3 from Lusaka Province and all 3 samples from Southern Province) were malaria positive by RDT but we did not have an estimated parasitemia by PET-PCR. Thus, these samples were, omitted from the last bloxplot on the right (pcr_pf_parasitemia). These details have been addressed in the Figure S3 legend.

Fig. S6 legend: SNP and sample missingness should be defined somewhere. Also there is a typo in the fourth line ('rom').

Response: Thank you for capturing this error. We corrected some typographical errors in the Fig. S5 legend (Fig. S6 in the original manuscript) . SNP and sample missingness are now defined in the revised manuscript at P. 5-6, L. 126-129 and in the Methods section.

Fig. S8 legend title: Figure does not contain information about allele frequencies.

Response: Correct. We removed “allele frequencies” in the title of Fig. S7 (corresponding to Fig. S6 in the original manuscript).

Fig. S14 legend is confusing: it talks about cM, but the actual plot is in kb and the implications of genetic distance don't seem to be stated anywhere.

Response: Thank you for the suggestion. We revised Fig. S11 (corresponding to Fig. S14 in the original manuscript) which now illustrates and contextualizes cM and kb. The legend has been modified to the following: Figure S11. Genetic distance and length (inset) of pairwise IBD segments across the 241 *P. falciparum* sequenced genomes. The genetic distances (measured in centimorgan [cM]) are calculated by assuming a constant recombination rate across the *P. falciparum* genome (13.5Kb/cM). The x-axis represents the length of IBD genomic segments in cent Morgan (cM) and the y-axis represent their frequency across all samples. The vertical dashed red line represents the median genetic distance, equivalent to 8 cM, which corresponds to approximately to six generations (range= 3-239cM).

Reviewer #2 (Remarks to the Author):

In this manuscript, Fola and colleagues describe the genomic diversity and relatedness of Plasmodium falciparum parasites sampled across Zambia. The approach is not novel, but the

data set is new and provides previously unavailable insight into parasite dynamics in Zambia. Genomic studies of this vein establish critical baseline understandings that can be used later to assess the efficacy of disease interventions. I offer a few points to make the analyses more robust, but overall, I found the approach thorough and appropriate.

IBD analysis:

- While isoRelate can accommodate multiclonal infections with a diploid model, its power depends on the number of genomes and the genome ratios. As a check to ensure the data are within reasonable bounds, the authors could compare the IBD patterns obtained with monoclonal-only vs polyclonal samples.

Response: We appreciate the reviewer raising this important question. Polygenomic infections are an inherent challenge when working with pathogen genomics, and we used isoRelate by purpose as this is the only currently available software that accommodates polygenomic infections without phasing information with a diploid model for IBD analysis. According to isoRelate power analyses, the power of detecting IBD segments is highest for monoclonal samples and progressively decreases for high MOIs. The accuracy of detected IBD segments is close to 1 for all MOI statuses as long as the size of IBD segments is larger than 3cM (see Fig2 in Henden et al. 2018). In our analyses, we required the segments to be at least 50Kb (~3.7 cM) long with at least 20 SNPs, which should be a conservative measure for detecting IBDs, especially in polyclonal samples. Since monoclonal samples only comprise 20% of the Zambian parasite samples, including all the samples in the analyses greatly increases the power of detecting regions under selection.

We demonstrate this by computing X_{iR} separately for monoclonal-only and polyclonal-only samples and inferred significant regions/loci again. As shown in Fig R3, monoclonal samples alone only detect signals on chromosomes 6 and 12, while polyclonal samples pick up different signals. The differences are not only due to the reduced power with fewer samples in the monoclonal group but also represent a geographic bias, as demonstrated in Figure 5 and Figure S12. 36% of western provinces' samples are monoclonal, while only 15% of eastern provinces' samples are polyclonal. Therefore, selection signals in monoclonal samples are biased towards western provinces (Figure S12).

Fig R3. Comparison of significantly selected SNPs and regions using all the samples vs. monoclonal or polyclonal samples alone.

In general, isoRelate is less affected by polyclonal infections according to the original paper (<https://pubmed.ncbi.nlm.nih.gov/29791438/>) and several follow-up studies that used isoRelate for IBD and selection signal estimations of *P. falciparum* from global WGS data set (<https://pubmed.ncbi.nlm.nih.gov/31439796/>) and *P. vivax* (<https://www.ncbi.nlm.nih.gov/pmc/articles/PMC7425983/>). Also, in the present study we captured selection signals previously identified in African *P. falciparum* populations suggesting a robust estimation of IBD metrics by isoRelates.

- *Precise language isn't consistently used. For instance: "only 3.21% (906/28,203) of pairs of genomes displaying at least one block of IBD shared". Technically this is pairs of *samples*, many of which contain multiple genomes. This may appear to be splitting hairs, however, such differences will be important later if this study is used as a baseline for understanding future epidemiological patterns.*

Response: Thanks for your suggestions. This and other wording ambiguities have been edited throughout the manuscript.

Selection analysis:

- *As with the IBD analysis, I would advise more exact language in regards to the selection analysis. These are all *putative* or *potential* signatures of selection. At times this more circumspect language is dropped.*

Response: We added *potential* signatures of selection as suggested in the relative sections in the revised manuscript.

- *It is worth noting that the isoRelate authors suggests coupling its selection test with other selection tests to obtain the most robust results,*

Response: We appreciate this observation and agree that coupling IsoRelate with another selection test will make our selection results more confident. We received a similar comment from reviewer #1 and addressed it accordingly. To ensure the credibility of IBD analyses, we also used another algorithm, integrated haplotype score (iHS), for inferring selection. iHS relies on phased data, we therefore restricted our analyses to monoclonal samples. We identified selected regions that contain at least two extreme markers with |iHS| values above 5% genomic significance level and found similar selection patterns suggesting selection results from isoRelate are robust (see Fig. R4). Common regions between the two methods include chromosomes 6 and 8. However, iHS looks for extended haplotypes without a minimum required haplotype lengths in contrast to IBD. Thus, iHS is better at identifying older selection as the gene list contains many more surface antigens, while isoRelate aims to detect selection within 200 generations (which correspond to the last 60 years of parasite evolution).

Figure R4. Comparison of significantly selected SNPs and regions using X_{iR} vs. Integrated Haplotype Score (iHS) across 14 *P. falciparum* chromosomes. The graph shows $-\log_{10}$ transformed p-values per biallelic SNPs. The shaded areas represent regions with at least two significantly selected SNPs within a window size of 50Kb. Overlapping areas are joined together. Note that X_{iR} analyses include all 241 samples (A), while iHS is performed on the 50 monogenomic samples only (B). The plot was generated using the R package rehh.

- Can anything more be said about the siblings and clones? This struck me as one result that might have implications for understanding local transmission and could potentially be explored further.

Response: We appreciate the reviewer's suggestion. In the revised manuscript we have now significantly revised the IBD-based relatedness analyses conducted on all 241 WGS sequenced samples compared to 238 samples (the 241 data set included the 238 samples originating from the well-represented seven out of the ten provinces, and 3 samples from three provinces with very low transmission). We identified 23 parasite pairs related at the level of second cousins and above. A few clusters of highly related parasites sharing

their genome IBD >50% and >90% were identified, including 3 half siblings (MOI>1) and 8 clonal lineages (MOI=1)/pairs that shared one clonal lineage (MOI>=1), respectively (Figure 4A). Most of these highly related parasite pairs were identified within the same cluster and province, providing evidence of local transmission, and inbreeding in six provinces (Fig. 4B). Interestingly, we identified only one instance of long-distance clonal sharing between non-neighboring provinces (Luapula and Southern Provinces) (Figure 4B) (P. 8, L. 179-201).

- The authors posit that there may be different selection pressures across the country. I think they could additionally explore other hypotheses like local patterns of relatedness and differences in power.

Response: Thank you for the insightful suggestion. At the regional level, we observed and reported variation in genomic regions under selection between eastern and western provinces, which constitute two transmission zones in Zambia (Figure 5B, C, Figure S12)(P .10-11) and suggest that parasites experience different selection pressures due to exposure to different control interventions, mosquito vectors, and environmental conditions. The sample size variation per province and the small sample sizes for some provinces limit an in-depth analysis and comparison of local patterns of relatedness and other IBD metrics at the provincial level.

-To my eyes, the signals picked up by the XiR approach agree with previous selection scans in Africa. Are there any loci that are novel? Are there any loci that are likely artifacts?

Response: We have now significantly revised the selection signals from IBD analyses and contextualized the signals picked up by the XiR approach in the Zambian parasite populations with existing literature. Specifically, the overall selection pattern in Zambia resembles the positive selection signature from IBD analyses of Malawian genomes (Figure 6 in Henden et al.21) as well as pyrimethamine-associated selection signal from association studies in Senegal (Figure 3 in Park et al.28). Further comparisons of loci under selection with previous evidence are provided in the revised manuscript in the Results section, P. 9-11.

PCA:

- The PCA would be more robust if highly related samples (ie, clonal or nearly clonal) are pruned. Otherwise, these pairs can artificially drive the partitioning. This doesn't appear to be an issue with the results, but it would still be best practices to do this.

- How is polyclonality handled in the PCA

Fig R5. PCA with monoclonal samples excluded.

Response: Thank you for your insights. We addressed the comments accordingly. The PCA analysis was performed including only polyclonal samples (Fig. R5) to check whether monoclonal samples may obscure other potential clustering that could be observed. However, we didn't find new evidence of population structure from the PCA using polygenomic infections alone (Fig. R5). For the PCA analysis we used the filtered SNP VCF files as input file as described in the Methods and no additional filtering was done. We used the R package SNPRelate to conduct the PCA, which calculates the genetic covariance matrix from genotypes and computes the correlation coefficients between sample loadings and genotypes for each SNP and which has been commonly used in malaria studies:

<https://journals.plos.org/plosntds/article?id=10.1371/journal.pntd.0008962>,
<https://www.ncbi.nlm.nih.gov/pmc/articles/PMC9576839/>, and
<https://pubmed.ncbi.nlm.nih.gov/35007277/>).

Other small edits:

- It could help to define "cluster" early in the manuscript. Since this isn't a term I usually associate with sampling location, I was initially confused.

Response: Thank you for your suggestions. We have sought to more clearly define what a cluster is in the revised manuscript but rely on reference to the source publication for additional details (<https://www.path.org/resources/zambia-natl-malaria-indicator-survey-mis-2018/>).

- On page 8 (top), the text reads "Mean IBD within clusters showed a strong negative correlation with inter-cluster geographic distance (Figure 4D)". I assume this should be: "Mean IBD *between* clusters..."

Response: We thank the reviewer for noting this detail. Yes, we meant "Mean IBD between clusters". However, we found an error in the original pairwise IBD vs. geographic distance analyses and redid the analyses. We found IBD above 5% only at the local cluster level with only two examples of long-distance sharing: one within the province, and one between two provinces. The results show that high IBD sharing is almost exclusively limited to very short distances. We, therefore, removed the original Fig 4D.

Reviewer #3 (Remarks to the Author):

This paper is very well written and it highlights novelty in the area of parasite genomics. The paper highlights some interesting findings. The methods section is clearly written and concise, and the results are also very well written though overall, it would have been nice for the authors to highlight further on the drug resistance markers that were identified as these seem to key in the fight against malaria and they would also be very good for control recommendations.

Response: We are appreciative of the reviewer's positive comments and relevant question. As the reviewer noted, this study and its findings are of great importance to Zambia malaria control and elimination efforts. In addition to the ability to infer transmission dynamics from parasite genomic data, the power of *P. falciparum* WGS and thus of this WGS data set from Zambia is that antimalarial drug resistance profiles of known and candidate genetic markers based on current knowledge and molecular mechanisms can be mined. We have now significantly revised the selection signals from IBD analyses and expanded on the discussions on the selected sites related to drug resistance, especially SP. We also compared the genomic selection patterns to those of other African and Asian countries (P. 9-11). Lastly, we also analyzed copy number variation (CNV) of selected genes and found evidence of CNV for several markers important for drug resistance (P. 15, L. 366-377). We believe, with these new additions to the Results, the study provides important guidance to malariologists as well as policymakers.

REVIEWERS' COMMENTS:

Reviewer #1 (Remarks to the Author):

The authors have successfully addressed the technical issue I raised and now recommend publication.

Reviewer #2 (Remarks to the Author):

In this revision, Fola and colleagues have strengthened their manuscript. As I wrote earlier, the work remains a useful contribution to our understanding of parasite dynamics in Zambia. The analysis is appropriate and well documented.

I am still not fully convinced by the interpretation that all the XiR outliers reflect selection. Like reviewer 1, I am concerned that SNP density may contribute to the signal, and the provided plots do not fully address this concern. Similarly, I find it telling that the iHS and XiR results diverge in several regions of the genome. However, I do not feel this should hinder publication of the analysis. The authors correctly point out that their approach has precedents in the field, and so while I would advocate for a more circumspect treatment of the data (and a more rigorous assessment of potential artifacts), I recognize that this is a debate for the field to have openly and more broadly.

I would, however, strongly encourage the authors to include the plots from the rebuttal letter (eg, SNP density, iHS) in the supplement so that readers can access this information and freely interpret it.

REVIEWERS' COMMENTS:

Reviewer #1 (Remarks to the Author):

The authors have successfully addressed the technical issue I raised and now recommend publication.

Reviewer #2 (Remarks to the Author):

In this revision, Fola and colleagues have strengthened their manuscript. As I wrote earlier, the work remains a useful contribution to our understanding of parasite dynamics in Zambia. The analysis is appropriate and well documented.

I am still not fully convinced by the interpretation that all the XiR outliers reflect selection. Like reviewer 1, I am concerned that SNP density may contribute to the signal, and the provided plots do not fully address this concern. Similarly, I find it telling that the iHS and XiR results diverge in several regions of the genome. However, I do not feel this should hinder publication of the analysis. The authors correctly point out that their approach has precedents in the field, and so while I would advocate for a more circumspect treatment of the data (and a more rigorous assessment of potential artifacts), I recognize that this is a debate for the field to have openly and more broadly.

I would, however, strongly encourage the authors to include the plots from the rebuttal letter (eg, SNP density, iHS) in the supplement so that readers can access this information and freely interpret it.

Response: We appreciate these valuable suggestions. The SNP density plots and discussion on the relationship between SNP density and XiR selection signals have been incorporated (Supplementary Figure 14 and Discussion section P.16-17, L. 407-417). Furthermore, we included the comparative analysis of selected SNPs and regions using XiR and the standardized integrated haplotype score (iHS), revealing somewhat similar selection patterns, particularly on chromosomes 6 and 8. We discussed the potential reasons for observed discrepancies, attributing them to the distinct detection capabilities of the two methods (Supplementary Figure 13 and Results section P. 13, L. 313-321). The methods for the above analyses are reported In Methods, P. 24, L. 603-611.